# A Novel Splice Variant of the Inhibitor of Growth 3 Lacks the Plant Homeodomain and Regulates Epithelial–Mesenchymal Transition in Prostate Cancer Cells

**DOI:** 10.3390/biom11081152

**Published:** 2021-08-04

**Authors:** Anna Melekhova, Mirjam Leeder, Thanakorn Pungsrinont, Tim Schmäche, Julia Kallenbach, Marzieh Ehsani, Kimia Mirzakhani, Seyed Mohammad Mahdi Rasa, Francesco Neri, Aria Baniahmad

**Affiliations:** 1Institute of Human Genetics, Jena University Hospital, 07740 Jena, Germany; Anna.melekhova@uni-jena.de (A.M.); thanakorn.pungsrinont@med.uni-jena.de (T.P.); tim.schmaeche@uniklinikum-dresden.de (T.S.); julia.kallenbach@uni-jena.de (J.K.); Marzieh.ehsani@med.uni-jena.de (M.E.); kimia.mirzakhani@uni-jena.de (K.M.); 2Department of Adult and Pediatric Urology, Jena University Hospital, 07743 Jena, Germany; Mirjam.danner@med.uni-jena.de; 3Department of Visceral, Thoracic and Vascular Surgery, University Hospital Carl Gustav Carus, Technische Universität Dresden, 01307 Dresden, Germany; 4National Center for Tumor Diseases (NCT/UCC), 01307 Dresden, Germany; German Cancer Research Center (DKFZ), Heidelberg, Germany; Faculty of Medicine and University Hospital Carl Gustav Carus, Technische Universität Dresden, Dresden, Germany; Helmholtz-Zentrum Dresden-Rossendorf (HZDR), Dresden, Germany; 5Leibniz Institute on Aging, 07745 Jena, Germany; mahdi.rasa@leibniz-fli.de (S.M.M.R.); francesco.neri@leibniz-fli.de (F.N.)

**Keywords:** prostate cancer, plant homeodomain, inhibitor of growth 3

## Abstract

Inhibitor of growth 3 (ING3) is one of five members of the ING tumour suppressor family, characterized by a highly conserved plant homeodomain (PHD) as a reader of the histone mark H3K4me3. ING3 was reported to act as a tumour suppressor in many different cancer types to regulate apoptosis. On the other hand, ING3 levels positively correlate with poor survival prognosis of prostate cancer (PCa) patients. In PCa cells, ING3 acts rather as an androgen receptor (AR) co-activator and harbours oncogenic properties in PCa. Here, we show the identification of a novel ING3 splice variant in both the human PCa cell line LNCaP and in human PCa patient specimen. The novel ING3 splice variant lacks exon 11, ING3∆ex11, which results in deletion of the PHD, providing a unique opportunity to analyse functionally the PHD of ING3 by a natural splice variant. Functionally, overexpression of ING3Δex11 induced morphological changes of LNCaP-derived 3D spheroids with generation of lumen and pore-like structures within spheroids. Since these structures are an indicator of epithelial–mesenchymal transition (EMT), key regulatory factors and markers for EMT were analysed. The data suggest that in contrast to ING3, ING3Δex11 specifically modulates the expression of key EMT-regulating upstream transcription factors and induces the expression of EMT markers, indicating that the PHD of ING3 inhibits EMT. In line with this, ING3 knockdown also induced the expression of EMT markers, confirming the impact of ING3 on EMT regulation. Further, ING3 knockdown induced cellular senescence via a pathway leading to cell cycle arrest, indicating an oncogenic role for ING3 in PCa. Thus, the data suggest that the ING3Δex11 splice variant lacking functional PHD exhibits oncogenic characteristics through triggering EMT in PCa cells.

## 1. Introduction

The inhibitor of growth (ING) gene family compromises five protein-coding members, *ING1* to *ING5*, and the pseudogene *INGX*. They were identified based on sequence homology to *ING1*, which was discovered first [1]. Interestingly, all members are highly evolutionary conserved among different species [2,3], which indicates their fundamental role in biological processes. The ING3 amino acid sequence of mice and human share 94.5% identity. 

Except for ING5, some splice variants are known for ING family members, providing additional complexity to this family [4]. As the name of this family implies, ING proteins were reported to play a role in inhibition of cellular proliferation via induction of cell cycle arrest or apoptosis [1,5,6,7]. Indeed, ING members functionally act as type II tumour suppressors in many cancer types. They were found to be downregulated in various malignancies, which correlates with poor prognosis for patient survival [8,9,10,11,12]. Additionally, ING factors are involved in the regulation of various cancer hallmarks as metastasis, migration, angiogenesis, and DNA damage response [6,13,14,15].

The plant homeodomain (PHD) is a highly conserved motif within the ING family members. The PHD interacts with the histone modifications of histone H3 at lysine 4, H3K4me, H3K4me2, and H3K4me3 with increasing affinity to the higher methylation state [16]. By binding to histones and epigenetic regulators with different domains, ING members change the epigenetic status of the chromatin by recruiting histone deacetylases or histone acetyltransferases [17]. In contrast to ING1 and ING2 that act as corepressors of the androgen receptor (AR), evidence suggests that ING3 acts as a co-activator of AR [18,19,20]. Thus, it is suggested that ING3 acts as an oncogene in prostate cancer (PCa). ING3 is frequently downregulated in different cancer types, such as head and neck carcinoma, melanoma, and hepatocellular carcinoma [11,21,22,23]. In contrast, ING3 expression positively correlates with AR-activity, poor prognosis, and metastatic potential of PCa, thus supporting an oncogenic role of ING3 in PCa [20]. Of note, ING3 is the only ING-family member known that leads to early embryonic lethality at day 10.5 upon its homozygous knockout in mice, as its expression in embryo’s brain is required for ectoderm differentiation and prenatal formation of the brain [24]. This indicates that ING3 might be involved in differentiation and also cell migration or EMT. Furthermore, ING3 is crucial for asymmetrical cell division and apoptosis induction [5,25].

Here we show the identification of a novel ING3 splice variant (ING3Δex11) lacking the PHD in both human PCa cell line and patient specimen obtained from prostatectomy in vivo. Therefore, the exon 11 deletion enables us to acquire functional insights into the PHD activity of ING3. Overexpression and knockdown experiments in monolayer cell culture and 3D human PCa spheroids indicate that the PHD of ING3 regulates EMT factors and the PHD-mediated inhibition of EMT. These data support an oncogenic role of ING3Δex11 splice variant in PCa cells.

## 2. Results

### 2.1. Identification and Functional Analysis of a Novel PHD-Lacking ING3 Splice Variant

Using different sets of primers to perform reverse transcription PCR of total RNA from the human PCa cell line LNCaP, we identified an unexpected additional amplified PCR product (Figure 1A,B). Sequencing indicated that the smaller PCR products lack the exon 11, indicating an alternative splice variant of ING3, termed ING3∆ex11 (Figure 1C). The skipping of exon 11 causes an in-frame deletion of 39 nucleotides resulting in loss of 13 amino acids within the highly conserved PHD of the ING family members (Appendix A). Since ING3∆ex11 was not found in the NCBI database, we analysed several native patient tumour samples obtained after prostatectomy for the expression of this splice variant. We identified the splice variant in all tumour samples (Figure 1D). The expression level of ING3∆ex11 in patient samples was significantly lower compared with all ING3 transcripts. Structure predictions of full-length ING3 by the Protein Homology/analogY Recognition Engine (PHYRE2) software indicates that the majority of more than 62% of both ING3 and ING3∆ex11 is predicted to have an intrinsic disordered structure. Therefore, we focused on the carboxyterminal region of ING3, which harbours the PHD. The prediction using Expasy SWISS-MODEL indicates the lack of Zn^2+^ ion-binding property of the truncated PHD of ING3∆ex11 and a changed tertiary structure of PHD motif of ING3∆ex11 by Expasy SWISS-MODEL (Figure 1E). A similar changed tertiary structure of ING3∆ex11 was predicted by the PHYRE2 software (Appendix A). 

To analyse the cellular function of the novel ING3∆ex11 splice variant, overexpression experiments were performed using LNCaP cells (Figure 2A). Growth assays indicate a growth inhibition of LNCaP cells by the ING3∆ex11 splice variant (Figure 2B). Moreover, we observed massive detachment of cells specifically from those plates transfected with ING3∆ex11 expression vector leading to fewer adherent cells. Since ING3 can induce apoptosis [26], the generation of cleaved-PARP (c-PARP) was determined. We identified also an ING3 isoform with an alanine to threonine conversion (ING3 A222T) that did not show a difference compared with ING3 and therefore used initially as a further control. c-PARP was not detected as well as cleaved caspase 3 (Figure 2A), indicating that apoptosis is not a reason for the massive detachment as well as it not being a relevant pathway for the observed growth reduction.

Therefore, we analysed for the induction of cellular senescence, another pathway for growth reduction. Interestingly, overexpression of full-length ING3 induced the senescence-associated β-galactosidase (SA-β-Gal) positive cells indicating the induction of cellular senescence by ING3, albeit significantly weaker by ING3∆ex11 (Figure 2C). In line with this, an induction of *CDKN**1A* encoding the cell cycle inhibitor p21 was observed both at protein and mRNA level (Figure 2D,E). Additionally, the expression of the cell cycle inhibitor *CDKN2A* encoding p16 was enhanced (Appendix A). However, based on the similarities between ING3 and ING3∆ex11, it indicates that the induction of cellular senescence might not be the only underlying mechanisms of ING3∆ex11-mediated growth reduction. Since both the AR antagonist enzalutamide (Enz) or supraphysiological androgen level induce cellular senescence [27,28], these ligands were used to analyse whether the ING3 splice variant affects the AR-mediated pathway to induce cellular senescence. As expected, treatment of LNCaP with these AR ligands mediated induction of cellular senescence, but neither ING3 nor ING3∆ex11 exhibited a detectable difference in the level of ligand-induced cellular senescence (Figure 2F). Taken together, besides an overlapping activity with ING3 in induction of cellular senescence, ING3∆ex11 specifically reduces number of adherent PCa cells by detachment. The data also indicate that the expression of a novel ING3 splice variant, ING3∆ex11, with deletion in the highly conserved PHD, can serve as a tool to understand the functional PHD activity.

### 2.2. ING3∆ex11 Induces EMT

Next, we generated 3D spheroids from transfected LNCaP cells (Figure 3) because a spheroid model mimics a tumour better in terms of complexity and drug delivery compared with monolayer cultures [29]. No obvious difference in the spheroid volumes were detected (data not shown). However, stained spheroid slices revealed morphological differences. We observed the formation of pore-like or lumen-like structures specifically inside those spheroids transduced with ING3Δex11 expression vector (Figure 3A). Such pore-like structures are an indicator of decreased E-cadherin expression in spheroids [30]. Therefore, it was hypothesized that ING3∆ex11 regulates the expression of EMT markers, as the induction of EMT might also cause the observed cell detachment. 

To address this hypothesis of regulation of EMT by ING3∆ex11, epithelial and mesenchymal markers as well as EMT-mediating transcription factors were analysed at both mRNA and protein level. Indeed, the expression of some key regulators of EMT are modulated specifically by ING3∆ex11 (Figure 3B). Whereas the expression of SNAI2, encoding Slug, and TWIST1 mRNA, was not affected by overexpression of ING3, it was enhanced by ING3∆ex11. The expression of ZEB1 was repressed by ING3; however, it was induced by ING3∆ex11. The mRNA level of SNAI1, encoding Snail, was slightly repressed by full-length ING3 and not affected by ING3∆ex11 (Figure 3B). These data indicate that compared with full-length ING3 with a functional PHD, ING3∆ex11 mediates specific transcriptional regulation of key regulatory EMT factors. 

Analysing EMT markers at the protein level, it shows induction of the mesenchymal markers N-cadherin and vimentin specifically by ING3∆ex11 expression (Figure 3C). The epithelial marker E-cadherin was also induced by ING3∆ex11, suggesting that at least a partial EMT is induced by ING3∆ex11, which is also observed for circulating tumour cells and cell migration [31]. Noticeably, ING3∆ex11 overexpression upregulates N-cadherin, which is slightly inhibited by full-length ING3 (Figure 3C), which may indicate a counteraction between ING3 and ING3∆ex11.

Thus, the data suggest that the lack of the PHD in ING3 regulates the expression of key EMT regulators and EMT markers. 

### 2.3. The Knockdown of ING3 Induces EMT

ING3∆ex11 overexpression revealed changes in expression of EMT markers at the protein level. Therefore, it was hypothesized that the full PHD motif of ING3 with functional PHD is required for protection against dysregulated EMT. To verify that the PHD of ING3 inhibits EMT, ING3 knockdown experiments were performed in order to analyse EMT markers. Since we observed that C4-2 cells have a higher ING3 level compared with LNCaP (Figure 4A), we used C4-2 cells for knockdown experiments. First, we analysed the effect of ING3 knockdown using two retroviral shING3 vectors, with the shING3(1) vector showing efficient ING3 knockdown at both mRNA and protein levels (Figure 4B,C). Therefore, shING3(2) and shLUC vectors were considered as experimental controls. Growth analysis indicates that the transduction of shING3(1) potently reduced cell growth (Figure 5A). This effect was observed for both transduced adherent cells (Figure 5A) as well as for spheroids generated from transduced shING3(1) cells (Figure 5B). Again, we did not observe enhanced apoptosis in line with undetectable apoptotic marker c-PARP (Figure 5C). Rather, we observed an enhanced level of cellular senescence associated with elevated p21 levels in transduced shING3(1) cells (Figure 5D,E), indicating that the cellular levels of ING3 are critical since both knockdown as well as overexpression induce cellular senescence.

Because EMT induction is suspected to occur rapidly after knockdown, it was decided to use efficient siRNA transfection and to perform protein extracts 72 h after transfection. The analysis of EMT markers indicate that the ING3 knockdown increases the mesenchymal markers N-cadherin and vimentin and decreases the epithelial marker E-cadherin as well (Figure 6). This supports the hypothesis that ING3 with functional PHD inhibits EMT. 

Since EMT is an important step towards metastasis, we analysed expression datasets of PCa patient specimens from TCGA. No data for the ING3 splice variant ING3∆ex11 were found in the TCGA datasets. Analysis of full-length ING3 expression levels suggests a slight upregulation in tumour samples compared with non-tumour samples (Figure 7A). Next, using the TCGA datasets, we analysed the ING3 expression with respect to the Gleason score, which is a measure of PCa aggressiveness, with an increased Gleason score being associated with higher metastatic potential [32]. Interestingly, dissecting the ING3 expression levels for each Gleason score, the data suggest an increase of ING3 mRNA levels up to Gleason score 8, whereas the samples with a higher Gleason score of 9 and 10 revealed a decline of ING3 expression (Figure 7B), indicating a lower ING3 expression in higher metastatic PCa. 

Taken together, a novel splice variant of ING3, ING3∆ex11, lacking a functional PHD, was identified, and its effect on cell proliferation and inhibition of EMT highlights the regulatory function of the PHD of ING3. 

## 3. Discussion

ING3 belongs to the ING family, which act as tumour suppressors, binding to the epigenetic histone mark of methylated histone H3 lysine 4 with their highly conserved PHD [6,33]. In general, ING family members positively regulate apoptosis and cellular senescence and negatively regulate proliferation, tumour progression, and metastatic potential of cells [4,33]. In line with this, ING3 was reported to have a tumour suppressive function in many different cancer types [5,11,23]. However, an oncogenic role of ING3 in PCa is suggested based on its AR co-activator function [20]. Here we revealed a dual role of ING3 in PCa cells. On the one hand, ING3 acts as an oncogene because its knockdown reduces PCa cell proliferation by triggering cellular senescence. Although the induction of cellular senescence is an irreversible cell cycle arrest, the senescence-associated secretory phenotype may promote cell growth of neighbouring cancer cells as part of a regeneration program. On the other hand, ING3 may function as a tumour suppressor by inhibiting EMT induction. One possibility is that the tumour suppressor activity of ING3 in other cancer types may derive from splice variants. Another possibility is that ING3 in PCa has a tumour suppressive function specifically for EMT. These data suggest the possibility that ING3 in PCa has a tumour suppressive function specifically for EMT but has an oncogenic role in cell proliferation.

Splice variants of ING3 have not yet been analysed in detail, and therefore, it is unclear what contributions full-length and ING3 splice variants have in cancer proliferation and EMT. An oncogenic role of ING3 in PCa cells may depend on the expression of ING3 as well as the ING3 splice variant, especially since ING3Δex11 can trigger EMT and overexpression of ING3 induces cellular senescence. Although cellular senescence induces cell cycle arrest, its role in the tumour microenvironment could transform the cancer cell niche into a growth-promoting tumour microenvironment by the senescence-associated secretory phenotype. Hypothetically, the tumour suppressor activity of ING3 in other cancer types may derive from splice variants as well. 

Focusing in detail on the analysis of ING3Δex11, the overexpression resulted in an enhanced expression of EMT-promoting transcription factors Slug, Twist, and Zeb1, which can reduce epithelial marker E-cadherin and induce mesenchymal state of cells. In addition, ING3Δex11 upregulated mesenchymal markers N-cadherin and vimentin at the protein level, confirming a cellular EMT process. Moreover, spheroids with ING3Δex11 overexpression formed lumen-like or pore-like structures. This morphology identified in PCa tumour spheroids indicates a decrease in cell–cell contact strength in spheroids and supports the notion of EMT induction. Interestingly, ING family members were reported to regulate EMT. For example, ING2 enhances EMT in renal proximal tubular cells and is likely to be involved in the development of a main pathological effect in diabetic kidney disease [34]. Both ING4 and ING5 suppress EMT in various cancer types [35,36,37,38]. 

Thus, the novel identified ING3 splice variant lacking the PHD suggests, that ING3 suppresses EMT promotion in a PHD-dependent manner, indicating that full-length ING3 could reduce EMT in PCa cells. Hence, tumour suppression of ING3 is presumably based on reducing metastatic potential. Analysing the ING3 expression dependent on the Gleason score, which is a measure of PCa aggressiveness, an increase of ING3 levels was connected with increasing Gleason score up to 7–8. However, at the higher Gleason scores of 9 and 10, which are characterized by increased metastatic potential, it is associated with decreased ING3 levels. This may indicate an enhanced metastatic potential by an increased EMT.

Our data suggest that newly identified splice variant ING3Δex11 counteracts ING3 full-length activity in regulating EMT factors. In contrast to overexpression of ING3, the novel splice variant ING3Δex11 triggers EMT in PCa cells, which was also observed by the knockdown of ING3. This suggests that the PHD motif protects for EMT and that lower ING3 levels promote EMT. Thus, the data suggest that the ING3Δex11 splice variant with the deletion of PHD exhibits oncogenic function through triggering EMT in PCa cells.

## 4. Materials and Methods

### 4.1. Cell Culture, Retroviral Transduction, and Transient Knockdown with siRNA

General procedures for culturing human PCa cell lines, LNCaP and C4-2, were previously described by Esmaeili et al. (2016) [2]. Transient transfection was performed using GeneJet according to manufacturer’s protocol. Retroviral gene transfer into PCa cells was performed as described earlier [2]. Retroviral transduction for stable knockdown was performed with vectors containing shRNA against ING3, whereas overexpression was performed with vectors containing ING3, ING3(A222T), and ING3Δex11 cDNA. Transient ING3 knockdown experiments were performed with pooled siRNA from Dharmacon, and as control, scrambled siRNA was used according to manufacturer’s protocol.

### 4.2. Generation and Analyses of 3D Spheroids

Spheroids were generated by forced floating method in 96-well ULA plates (1000 cells/well). AR ligands or DMSO diluted in fresh medium were added to spheroids 24 h after seeding. For each treatment, three technical replicates were conducted. The spheroids were incubated for indicated days, whereas the medium was refreshed every 72 h. For that purpose, half of the old medium was replaced by fresh medium containing AR ligands or DMSO. Spheroids were sliced using cryotome into 7 μm sections at −20 °C and blade temperature –30 °C. Sliced spheroids were carefully picked up by HistoBond+ adhesion-microscope slices (Marienfeld), dried at room temperature for 30 min, followed by counter-staining with haematoxylin.

### 4.3. Senescence Associated Beta-Galactosidase (SA β-Gal) Assays

Senescence assays of monolayer cells were performed as described previously [27]. To stain spheroids, they were fixed in 4% paraformaldehyde (PFA) and washed 3 times with 1X PBS, pH 6. The incubation with freshly prepared SA β-Gal staining solution was performed at 37 °C without CO_2_ for 48 h. X-Gal-stained spheroids were embedded in paraffin, and 4 µM cross-sections were prepared. After dehydration, the nuclei were counter-stained with haematoxylin (Roth, T865.1) for 2 min. The spheroid cross-sections were imaged using Axiolab microscope (Zeiss, Jena, Germany). The 1 nM R1881 and 10 µM enzalutamide were used to induce cellular senescence as described previously [2,28]; 0.1% DMSO served as solvent control.

### 4.4. Reverse Transcription Quantitative Real-Time PCR (qRT-PCR)

RNA extraction and qRT-PCR assays were performed as explained elsewhere [27]. The primer sequences are listed in Table 1, Table 2 and Table 3.

### 4.5. Antibodies and Western Blot Analyses

Preparation of whole cell lysates and Western blotting were performed as described elsewhere [27]. Antibodies raised against the following proteins were used: E-cadherin (Cell Signaling, 3195S), ING3 (Abclonal, A5832), N-cadherin (Cell Signaling, 13116S), p21 (Cell Signaling, 2947), PARP (Cell Signaling, 9546), caspase 3 (Cell Signaling, 9662), vimentin (Cell Signaling, 5741), and β-actin (Abcam, ab6276). As secondary antibody, horseradish peroxidase-conjugated anti-mouse (Cell Signaling, 7076), horseradish peroxidase-conjugated anti-rabbit (Cell Signaling, 7074), horseradish peroxidase-conjugated anti-mouse IgG (Santa Cruz, sc-2005), or anti-rabbit IgG (Santa Cruz, sc-2370) were used. The detection was performed by ImageQuantTM LAS 4000 (GE Healthcare Bio-Sciences AB). LabImage 1D software (Kapelan Bio-Imaging Solutions, Leipzig, Germany) was applied for quantification of protein of interest relative to the loading control (β-actin). 

### 4.6. Ex Vivo Treatment of PCa Samples from Patients

Patients with radical prostatectomies were described previously [27], with ethical approval (3286-11/11 and 2019-1502) conforming with the Declaration of Helsinki.

### 4.7. ING3 Structure Predictions

ING3 structure predictions were performed using PHYRE2—Protein Homology/analogY Recognition Engine (www.sbg.bio.ic.ac.uk/phyre2, accessed on 25 May 2021) and Expasy SWISS-MODEL (https://www.expasy.org/resources/swiss-model, accessed on 25 May 2021).

### 4.8. The Genome Cancer Atlas (TCGA) Database Analysis

The mapped results (bam file) of RNA sequencing of 456 tumour samples were downloaded from the TCGA (Project#24795) database (gdc.cancer.gov accessed on 6 April 2020). To calculate the FPKM (fragments per kilo base pair transcript per million reads), cuffdiff (v2.2.1) (http://cole-trapnell-lab.github.io/cufflinks/install/, accessed on 5 June 2014) was used with -library-norm-method quartile -total-hit-norm parameters with hg38 refseq annotation. The calculated FPKM for ING3 variant (NM-019071.3) was used for plotting. To compare ING3 expression in normal tissue (n = 50) with primary tumour (n = 50), paired samples (normal and tumour from the same patient) were used with a paired Wilcoxon two-sided test. For comparison between tumours with different Gleason scores, an unpaired Wilcoxon two-sided test was used.

### 4.9. Statistical Analysis

A two-tailed unpaired Student’s *t*-test was performed for statistical analysis using GraphPad Prism software. The calculations were made from the mean, standard deviation (SD), standard error of the mean (SEM), and number of replicates (n). A *p* value: *p* < 0.05 was considered as statistically significant (*) between two participant groups, *p* < 0.01 (**), *p* < 0.001 (***), and *p* < 0.0001 (****).

## Figures and Tables

**Figure 1 biomolecules-11-01152-f001:**
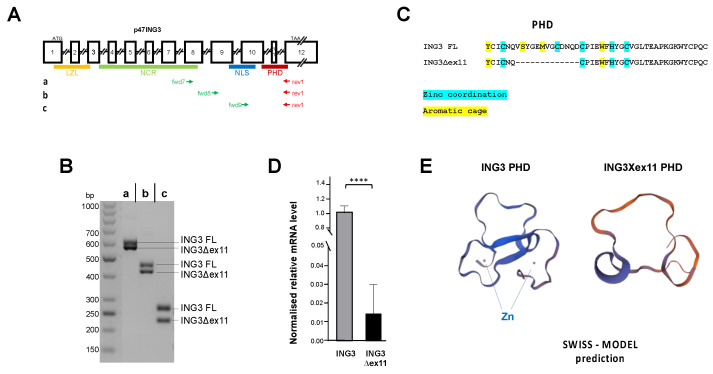
Identification of a novel ING3 splice variant lacking the PHD. Total mRNA from LNCaP cells were analysed with a set of primers for ING3 splice variants. (**A**) Intron exon structure of ING3 with the indicated protein domains, LZL, leucine zipper-like; NCR, novel conserved region; NLS, nuclear localisation signal; and PHD, plant homeodomain. (**B**) Agarose gel reveals two PCR products by each indicated primer set. (**C**) Cloning and sequencing of the lower migrating band indicates the lack of exon 11 of ING3 with an in-frame fusion between exon 10 and 12. (**D**) Patient samples (n = 6) obtained from prostatectomies were analysed for the presence of ING3 and ING3∆ex11 mRNA. One-tailed and two-tailed unpaired *t*-test showed same significance. (**E**) Predicted structure of ING3 and ING3∆ex11 protein by Expasy SWISS-MODEL indicates the lack of Zn^2+^ ion binding in the PHD and an aberrant folded PHD. *p* value < 0.0001 (****).

**Figure 2 biomolecules-11-01152-f002:**
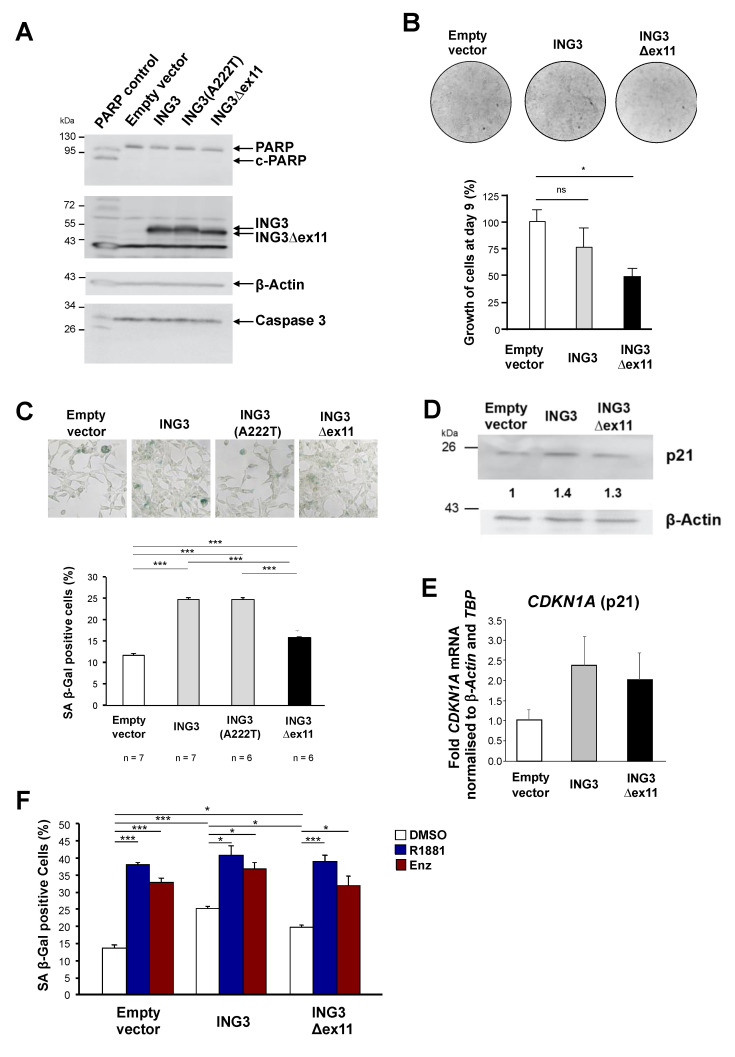
ING3∆ex11 expression inhibits cell growth and induces cellular senescence; LNCaP cells were transfected with indicated expression vectors for ING3, ING3∆ex11, or empty vector as control. (**A**) Western blot indicates the lack of apoptosis cleaved-PARP (c-PARP) and cleaved caspase 3. (**B**) Growth assays were analysed with crystal violet staining to indicate the inhibition of cell proliferation. (**C**) Cellular senescence was analysed by the senescence marker SA β-Gal activity. (**D**) Western blot indicates the upregulation of the cell cycle inhibitor p21. (**E**) qRT-PCR indicates p21 expression is upregulated by ING3 and ING3∆ex11 at mRNA level. (**F**) Analysis of SA β-Gal activity in the presence of androgen receptor ligands, Enz (enzalutamide) as AR antagonist, R1881 as an AR agonist, and DMSO as solvent control in cells transfected with the indicated expression vectors for ING3, ING3∆ex11, or empty vector as control. *p* values: *p* < 0.05 (*); *p* < 0.01 (**); *p* <0.001 (***); ns: not significant.

**Figure 3 biomolecules-11-01152-f003:**
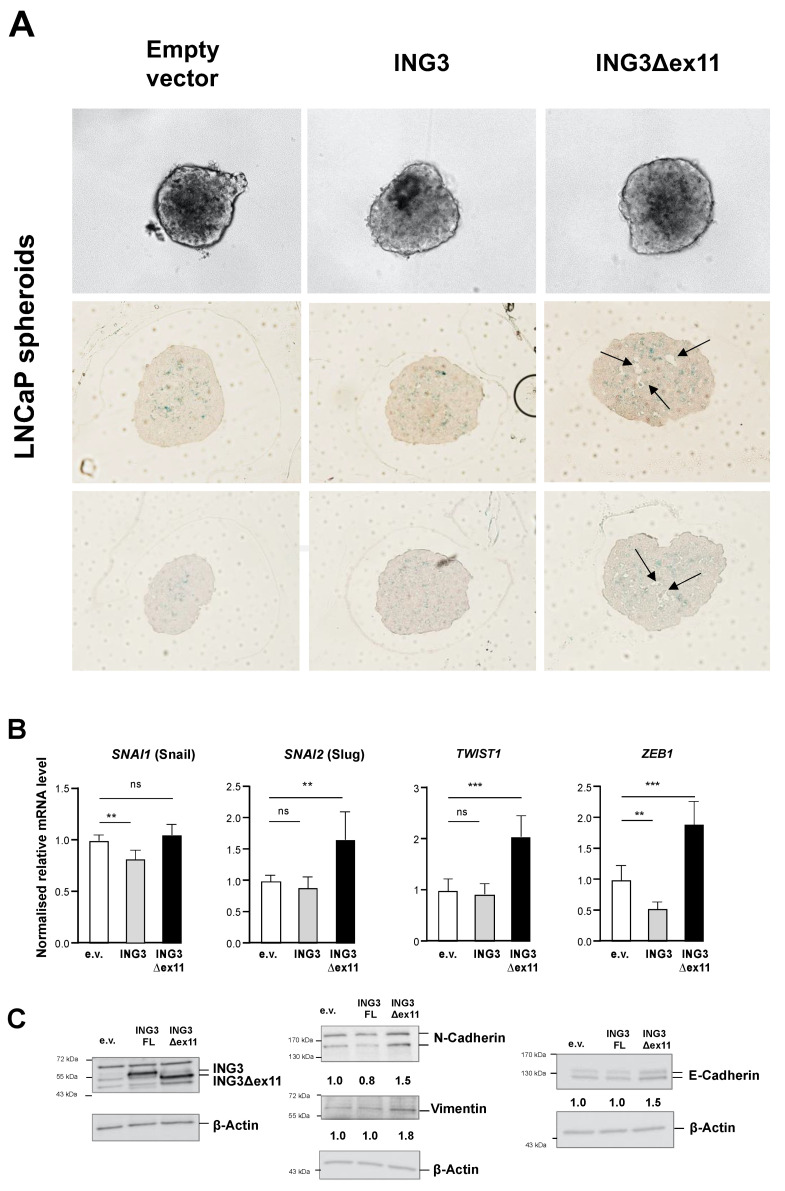
Formation of lumen and pore-like structures in ING3∆ex11 overexpressing LNCaP-derived tumour spheroids. (**A**) LNCaP spheroids were generated and sliced. Arrows indicate lumen and pore-like structures within spheroids. (**B**) Analysis of mRNA levels of key EMT regulators by qRT-PCR and (**C**) epithelial and mesenchymal markers at protein level by Western blot. *p* values: *p* < 0.05 (*); *p* < 0.01 (**); *p* <0.001 (***); ns: not significant; e.v., empty vector control.

**Figure 4 biomolecules-11-01152-f004:**
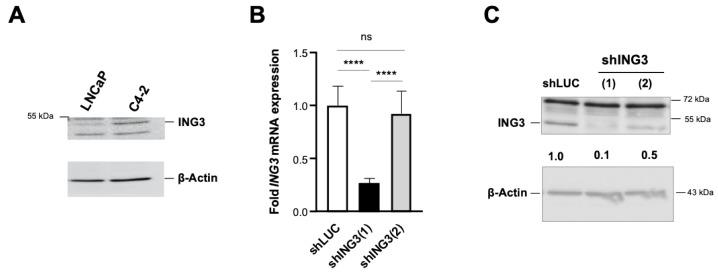
Knockdown of ING3 by retroviral transduction. (**A**) ING3 protein levels were compared between LNCaP and C4-2 cells by Western blotting. Short hairpin vectors were used to knockdown ING3 (shING3). Two vectors with different ING3 targeting sequences were used and transduced into ecotropic C4-2 cells. Seven days post-transduction knockdown, efficacy was analysed at mRNA (**B**) and protein (**C**) levels and compared with empty control vector shLUC, which was set arbitrarily as one. Results represent the mean of triplets, and error bars show standard deviation of the mean. *p* value < 0.0001 (****); ns: not significant.

**Figure 5 biomolecules-11-01152-f005:**
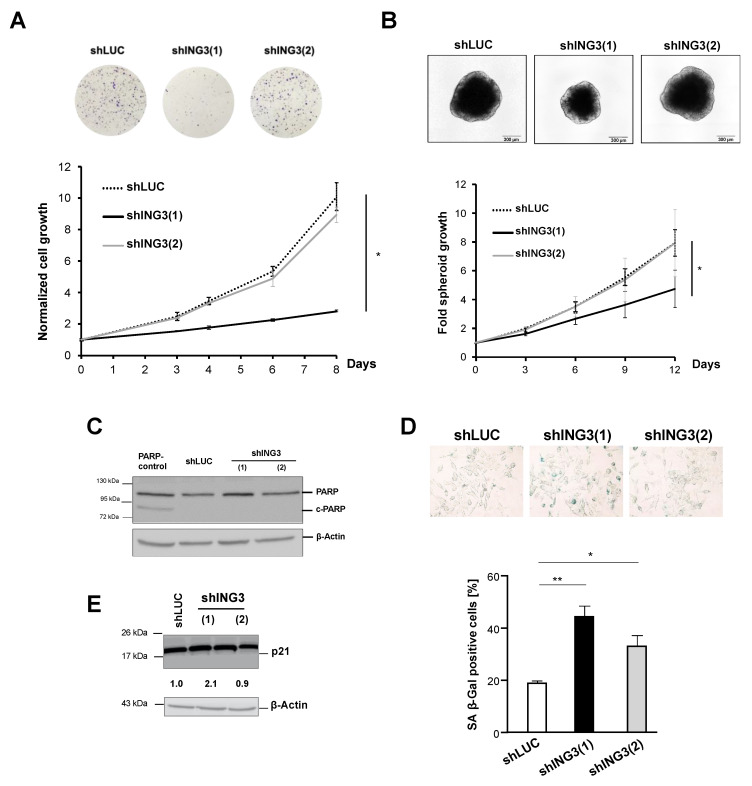
Knockdown of ING3 inhibits growth of adherent cells and tumour spheroids. (**A**) Transduced adherent C4-2 cells with the indicated vectors were analysed by staining with crystal violet at the indicated days. Top panel, pictures of dishes at day 8 of experiment. (**B**) Analysis of transduced C4-2 tumour spheroids. Top panel represents pictures of spheroids at day 12 of the experiment. Lower plot indicates the growth of spheroids at the indicated days. (**C**) Western blot analysis of the apoptotic marker cleaved PARP, c-PARP. (**D**) Detection of the senescence marker SA *β*-Gal activity in the transduced cells. (**E**) Western blot analysis of the cell cycle inhibitor p21. *p* values: *p* < 0.05 (*); *p* < 0.01 (**).

**Figure 6 biomolecules-11-01152-f006:**
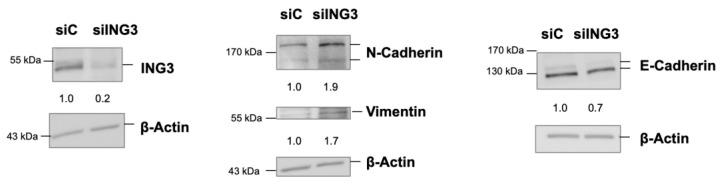
Knockdown of ING3 enhances the levels of EMT markers. Transient knockdown of ING3 by short interfering RNA (siING3) was used with scrambled siRNA as control (siC) to analyse key EMT markers by Western blotting.

**Figure 7 biomolecules-11-01152-f007:**
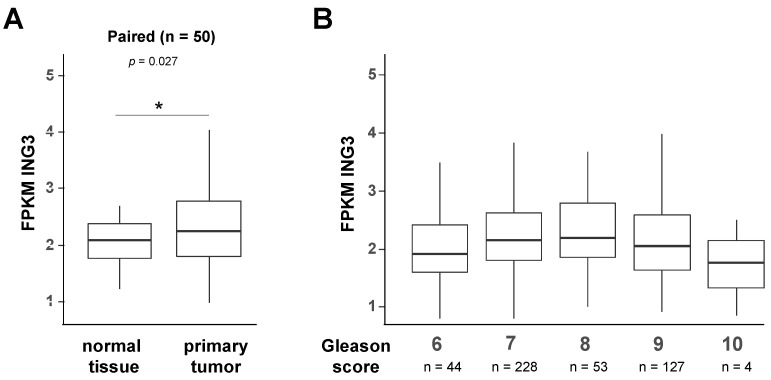
ING3 mRNA expression in PCa. TCGA database was analysed for full-length ING3 mRNA (NM-019071.3) levels in comparison (**A**) between primary tumour and normal tissues from the same patient and (**B**) among Gleason scores. Wilcoxon signed-rank test (paired for A and unpaired for (**B**) was performed to test the significance. Only significant comparisons are shown; *p* value < 0.05 (*).

**Table 1 biomolecules-11-01152-t001:** The list of primers used for gene expression analysis in qRT/PCR.

Gene (Protein)	Primer sequence 5′-3′
*CDKN1A* (p21)	fw: TCGACTTTGTCACCGAGACACCAC rev: CAGGTCCACATGGTCTTCCTCTG
*CDKN2A* (p16)	fw: CTTGCCTGGAAAGATACCG rev: CCCTCCTCTTTCTTCCTCC
*GAPDH 5*′	fw: GTGAACCATGAGAAGTATGACAAC rev: GAGTCCTTCCACGATACC
*ING3* (p47)	fw: CAGATGAAGGAGGGACGAAG rev: GCCGAAGATGATGAATAGCC
*ING3*Δ*ex11*	fw: GCATTTGTAATCAGTGCCCTAT rev: TCTGCTGCCTCTTCTCTTCATTGC
*SNAI1* (Snail)	fw: CAGTGCCTCGACCACTATGC rev: TGCTGGAAGGTAAACTCTGGAT
*SNAI2* (Slug)	fw: TTCGGACCCACACATTACCT rev: CTTCTCCCCCGTGTGAGTTCT
*TBP*	fw: GGCGTGTGAAGATAACCCAAGG rev: CGCTGGAACTCGTCTCACT
*a-Tubulin*	fw: TGGAACCCACAGTCATTGATGA rev: TGATCTCCTTGCCAATGGTGTA
*TWIST1* (Twist)	fw: CGGAGACCTAGATGTCATTGTTTC rev: CCCACGCCCTGTTTCTTTGAA
*ZEB1* (Zeb1)	fw: ACTCAACTACGGTCAGCCCT rev: ATCTTGTCTTTCATCCTGATTTCCA
*β-Actin*	fw: CACCACACCTTCTACAATGAGC rev: CACAGCCTGGATAGCAACG

**Table 2 biomolecules-11-01152-t002:** ING3 forward primers.

Primer	Localisation in Exon-Nr.	Sequence (5′-3′)	Localisation (bp)
fwd7	8	CCACAGCCTCTTCTAACAATG	709–729
fwd8	9	GGGACGAAGAACATCAAGT	857–875
fwd9	10	CTTCAAGCCAGCAGTCATC	1057–1075

**Table 3 biomolecules-11-01152-t003:** ING3 reverse primer.

Primer	Localisation in Exon-Nr.	Sequence (5′-3′)	Localisation (bp)
rev1	12	CTGTCAATCCAACGCAGC	1317–1300

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
