# Peer review of "A Novel Splice Variant of the Inhibitor of Growth 3 Lacks the Plant Homeodomain and Regulates Epithelial–Mesenchymal Transition in Prostate Cancer Cells"

_biomolecules, 2021, doi:10.3390/biom11081152_

Round 1
Reviewer 1 Report
The authors identified and partially characterised a novel isoform of ING3 gene in prostate cancer. This isoform lacks the PHD histone mark reader domain, but unlike the full-length ING3, induces an epithelial-to-mesenchymal transition.
Overall, very interesting results that are in agreement with oncogenic role of ING3 in PCa. However, I have a few issues with the data (or missing data...):
-Figure 3C is missing westerns for N-Cadherin and beta-actin (middle panel) and E-Cadherin and beta-actin (right panel).
-Figure 4C is missing signal for ING3 and beta-actin. There's a band at 55 kDa. Is it ING3? If it is ING3, the signal is not reduced by the shING3(1).
-Figure 5C, there's no signal for beta-actin...
-Figure 6, there's no signal for ING3, Vimentin, beta-actin, nor E-Cadherin.
-How was the A222T mutant found? Is it from a PCa patient?
-The induction of p21 is not striking (Figure 2D). Maybe look at mRNA or a different SA-betaGal marker.
-Could use TWIST as a positive control for induction of EMT. How does the TWIST-induced EMT compares with ING3delta11?
-The authors say that ING3delta11 induces specifically EMT genes. I would suggest to ChIP ING3 at EMT genes to support this claim.
-The authors claim that N-Cadherin is induced by silencing ING3 (Figure 6), but it is NOT the case. ImageJ analysis even suggest that the signal decreases from 143 to 133 units.
-A BioID approach of ING3 and ING3delta11 would help characterise this novel isoform and suggests mechanistically how ING3delta11 induces EMT.
Reviewer 2 Report
In the current study, authors were trying to identify a novel splice variant of ING3, which is lack of exon 11, from patient samples and LNCaP cell line. Also, authors were trying to show it functions in altering the ING3 regulation in EMT. There are many concerns need to be addressed before such conclusions can be made.
- In Fig 1D, authors showed the expression of total ING3 & ING3 Δex11 in patient samples, can authors show the expression of total ING3 & ING3 Δex11 in LNCaP, C4-2, VCaP cell lines? If the ING3 Δex11 expression is only a hundredth of total ING3, what is the significance of studying this splice variant?
- In Fig 2A, the ING3 Δex11 and ING3 overexpression models authors created should be similar to expression ration in the patient sample, which ING3 Δex11 expression should be in a hundredth of ING3 expression. The results could be artificial if ING3 Δex11 and ING3 are overexpressed equally. From the protein expression in LNCaP transfected with empty Vector showed the endogenous ING3 expression is very low, and it is hard to see the ING3 Δex11 expression that authors claimed detected in LNCaP. For the apoptosis marker, authors should detect cleaved-caspase 3 in addition to the total caspase 3. Can authors introduce ING3 A222T isoform, how it was discovered, what the function of this isoform. Can authors make a more detailed and scientific conclusion base on the data from figure 2, instead of quoted here “Taken together, the data indicate that ING3 Δex11 reduces number of adherent cells by detachment.”
- In Fig 3C, the western blot result in the middle and right had side part are invisible. Please remake the figure.
- In Fig 4A, there is no expression of ING3 Δex11 shown in the western blot as authors claimed. In Fig4C, although beta-actin bands are invisible, both shING3 bands showed bad knock-down efficiency. The subsequent results generated in the shING3 cell line are questionable.
- In Fig5A and B, shING3-1 and shING3-2 showed very different results, please include the third shING cell lines to repeat Fig 5A-E.
- In Fig6, except N-Cad protein expression, all the western blot results are invisible, please re-create all the western blot results.
- In Fig7, the total ING3 mRNA expression in the paired normal and primary tumor samples showed the increase of ING3 in tumor sample, can authors explain how it is against the knowledge of ING3 is a tumor suppressor?
Round 2
Reviewer 1 Report
There are still western blot panels missing in Figures 3-6.
Author Response
Thank you for pointing that out. However, on our screens, Mac or PC, or the printed version, we do see all Western blot panels. Therefore, it is hard for us to make the panels more clear.
Nevertheless, the Westernblot figures are now inserted in the mansucript as PNG data. Perhaps the tracking system interferes with the unvisible panels. So we uplodaed now the PDF file without track. Hopefully now the bands are clearly visible.

Reviewer 2 Report
Authors have done a great job answering the questions and concerns. This study will benefit the field greatly.
Author Response
Thank you very much.